# A universal test for gravitational decoherence

C. Pfister[1,2], J. Kaniewski[1,2], M. Tomamichel[2,3], A. Mantri[2], R. Schmucker[2], N. McMahon[4], G. Milburn[4] & S. Wehner[1]

Quantum mechanics and the theory of gravity are presently not compatible. A particular question is whether gravity causes decoherence. Several models for gravitational decoherence have been proposed, not all of which can be described quantum mechanically. Since quantum mechanics may need to be modified, one may question the use of quantum mechanics as a calculational tool to draw conclusions from the data of experiments concerning gravity. Here we propose a general method to estimate gravitational decoherence in an experiment that allows us to draw conclusions in any physical theory where the no-signalling principle holds, even if quantum mechanics needs to be modified. As an example, we propose a concrete experiment using optomechanics. Our work raises the interesting question whether other properties of nature could similarly be established from experimental observations alone—that is, without already having a rather well-formed theory of nature to make sense of experimental data.

[1] QuTech, Delft University of Technology, Lorentzweg 1, Delft 2628 CJ, The Netherlands. [2] Centre for Quantum Technologies, 3 Science Drive 2, Singapore 117543, Singapore. [3] School of Physics, University of Sydney, Sydney, New South Wales 2006, Australia. [4] ARC Centre for Engineered Quantum Systems, School of Mathematics and Physics, The University of Queensland, St Lucia, Queensland 4072, Australia. Correspondence and requests for materials should be addressed to S.W. (email: s.d.c.wehner@tudelft.nl).

Experiments[1–4] aiming at testing the presence—and amount—of gravitational decoherence generally go beyond established theory. Many theoretical models for gravitational decoherence have been proposed[5–25], and it is wide open if one of these proposals is correct. As such, experiments are of a highly exploratory nature, aiming to establish data points that constrain rival theoretical proposals. This task is made even more difficult by the fact that quantum mechanics and gravity do not go hand in hand, and indeed quantum mechanics may need to be modified in a yet unknown way in order to account for gravitational effects such as decoherence. We are thus compelled to design an experiment that provides a guiding light for the search for the right theoretical model—or indeed new physical theory—whose conclusions do not rely on quantum mechanics.

Here we propose an experimental procedure to estimate gravitational decoherence whose conclusions hold even if quantum mechanics would need to be modified. We first establish a general information-theoretic notion of decoherence which reduces to the standard measure within quantum mechanics. Second, drawing on ideas from quantum information, we propose a very general experiment that allows us to obtain a quantitative estimate of decoherence of any physical process for any physical theory satisfying only very mild conditions. Our method is fully general and could in principle be used to supplement many existing experimental proposals in a way that would allow us to draw conclusions from data even if quantum mechanics would need to be modified. Concretely, if a process (supposedly) causing gravitational decoherence can be probed experimentally, then our general method allows us to measure a parameter $\beta$ that translates into an upper bound on decoherence,

$$\text{Dec}(A|E) \leq h(\beta) \tag{1}$$

where $\text{Dec}(A|E)$ is the amount of decoherence of a system $A$ with respect to its environment $E$ (we will define this below). The function $h$ is plotted in the Discussion section for quantum mechanics, but also very general physical theories. As an example, we propose a concrete experiment using optomechanics to estimate gravitational decoherence in any such theory, including quantum mechanics as a special case. We note that our procedure could be used to probe any form of decoherence, but only in the case gravitational decoherence is there a pressing motivation for considering theories beyond quantum mechanics.

## Results

**Decoherence in quantum mechanics.** Before we turn to our general approach (see Fig. 1), let us first focus on the concept of decoherence within quantum mechanics as an easy warm-up. This demonstrates some principles that we will generalize to a broad framework of theories in the following section. Here we first show how the protocol given in Fig. 2 allows us to estimate quantum mechanical decoherence without knowing the decoherence process, and without doing quantum tomography to determine it. Traditionally, the presence of decoherence within quantum mechanics is related to the change of state due to measurement and the 'collapse of the wavefunction'. Decoherence is thereby often seen as a decay of the off-diagonal terms in the density operator $\rho$, corresponding to a (weak) measurement of the state. It is clear that this way of thinking about decoherence is entirely tied to the quantum mechanical matrix formalism, and also offers little in the way of quantifying the amount of decoherence in an operationally meaningful way.

The modern way of understanding decoherence in quantum mechanics in a quantitative way is provided by quantum information theory. One thereby thinks of a decoherence process as an interaction of a system $A'$ with an environment as described in Fig. 2, resulting in a quantum channel $\Gamma_{A' \to B}$. The amount of

decoherence can now be quantified by the channel's ability to transmit quantum information, that is, its quantum capacity (see Supplementary Note 1 for further background). For a finite number of channels, the relevant quantity is the single-shot capacity as determined by the so-called min-entropy $H_{\text{min}}(A|E)$[26,27].

Apart from its information-theoretic significance, the min-entropy has a beautiful operational interpretation that also makes its role as a decoherence measure intuitively apparent. Very roughly, the amount of decoherence can be understood as a measure of how correlated $E$ becomes with $A$. Suppose we start with a maximally entangled test state $\Phi_{AA'}$ where the decoherence process is applied to $A'$. This results in a state $|\Psi_{ABE}\rangle$ (see Fig. 2). If no decoherence occurs, the output state will be of the form $\Phi_{AB} \otimes |0\rangle\langle0|_E$ where $A' = B$. That is, $A$ and $B$ are maximally entangled, but $A$ and $E$ are completely uncorrelated. The strongest decoherence, however, produces an output state of the form $\Phi_{AE_1} \otimes \rho_{E_2} \otimes |0\rangle\langle0|_B$ where $A' = E_1$ and where $E$ is subdivided into subsystems $E = E_1 E_2$. That is, $A$ is now maximally entangled with $E_1$, whereas $A$ and $B$ are completely uncorrelated.

What about the intermediary regime? The min-entropy can be written as

$$H_{\text{min}}(A|E) = -\log(d_A \text{Dec}(A|E)) \tag{2}$$

where $d_A$ is the dimension of $A$, and (ref. 28)

$$\text{Dec}(A|E) = \max_{\mathcal{R}_{E \to A'}} F^2(\Phi_{AA'}, \mathbb{1}_A \otimes \mathcal{R}_{E \to A'}(\rho_{AE})) \tag{3}$$

and where $F$ denotes the fidelity

$$F(\rho, \sigma) = \text{Tr}\left(\sqrt{\sqrt{\rho}\sigma\sqrt{\rho}}\right). \tag{4}$$

The maximization above is taken over all quantum operations $\mathcal{R}_{E \to A'}$ on the system $E$, which aim to bring the state $\rho_{AE}$ as close as possible to the maximally entangled state $\Phi_{AA'}$ (see Fig. 3). Intuitively, $\text{Dec}(A|E)$ can thus be understood as a measure of how far the output $\rho_{AE}$ is from the setting of maximum decoherence (where $\rho_{AE} = \Phi_{AE}$ is the maximally entangled state). If there is no decoherence, we have $\rho_{AE} = \mathbb{1}/d_A \otimes \rho_E$ giving $\text{Dec}(A|E) = 1/d_A^2$ and $H_{\text{min}}(A|E) = \log d_A$. If there is maximum decoherence, we have $\rho_{AE_1} = \Phi_{AA'}$ giving $\text{Dec}(A|E) = 1$ and $H_{\text{min}}(A|E) = -\log d_A$ where $\mathcal{R}_{E \to A'} = \text{Tr}_{E_2}$ is simply the operation that discards the remainder of the environment $E_2$. A larger value of $\text{Dec}(A|E)$ thus

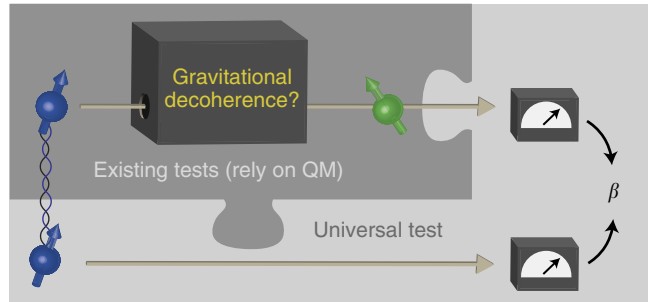

**Figure 1 | Illustration of our approach.** Our method can in principle be used in conjunction with any existing test for gravitational decoherence such that we can draw conclusions from the experimental data even if quantum mechanics would need to be modified. Intuitively, we combine a test that probes gravitational decoherence with a Bell test. From the estimated Bell violation $\beta$, we can draw quantitative conclusions about the amount of decoherence in any physical theory in which the no-signalling principle holds. The latter assumption could be relaxed further to theories that allow a small amount of signalling, in the sense that the no-signalling equation (10) is only satisfied approximately.

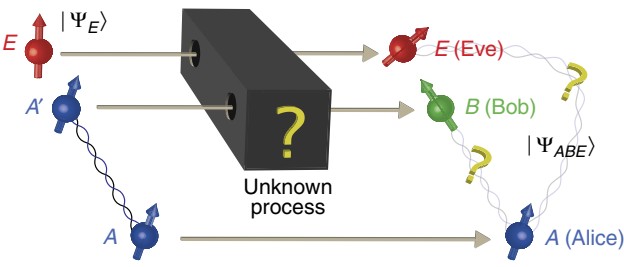

**Figure 2 | Diagram of the general setup.** A decoherence process—also known as a (quantum) channel—can be thought of as an interaction $U_I$ of the system $A'$ with an environment $E$. In quantum mechanics, the resulting state is the output of the channel $\rho_B = \Gamma_{A' \to B}(\rho_{A'}) = \mathrm{Tr}_E[U_I \rho_{A'} \otimes |\Psi_E\rangle\langle\Psi_E|U_I^\dagger]$. In general, $B$ (Bob) may be a smaller or larger system than $A'$. In the examples below, however, we will focus on the case where $A'$ and $B$ have the same dimension, corresponding to the case where a fixed system $A' = B$ experiences some interaction with another system $E$ (Eve). The channel's (in)ability to preserve quantum information—and therefore the amount of decoherence—can be characterized by how well it preserves entanglement between an outside system $A$ and $A'$. We note that our treatment of theories that go beyond standard quantum mechanics makes no statement whether the environment is an actual physical system, or merely a mathematical Gedanken experiment possibly used to describe an intrinsic decoherence process. In full generality, the experiment consists of a Bell experiment in which a source of decoherence is introduced deliberately. For simplicity, we consider an experiment for the CHSH inequality, although our analysis could easily be extended to any other Bell inequality. In each run, a source prepares the maximally entangled state $\Phi_{AA'}$, where $A'$ is subsequently exposed to the decoherence process to be tested. We then perform the standard CHSH measurements: system $A$ is measured with probability 1/2 using observables $A_0 = \sigma_X$ and $A_1 = \sigma_Z$ respectively. System $B$ is measured using observables $B_0 = (\sigma_X - \sigma_Z)/\sqrt{2}$ and $B_1 = (\sigma_X + \sigma_Z)/\sqrt{2}$ with probability 1/2 each. Performing the experiment many times allows an estimate of $\beta = \mathrm{Tr}[\rho_{AB}(A_0 \otimes B_0 + A_0 \otimes B_1 + A_1 \otimes B_0 - A_1 \otimes B_1)]$.

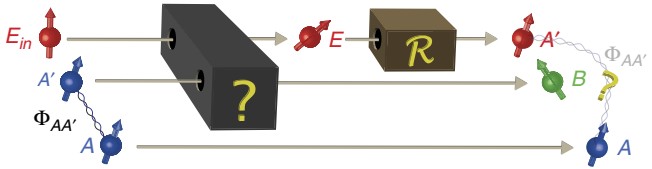

**Figure 3 | Intuitive picture of the decoherence quantity.** After the decoherence process, Eve (who controls the environment) performs an operation $\mathcal{R}_{E \to A'}$ in order to reach a state that is as entangled with system $A$ as possible. The decoherence quantity is a measure for how close Eve can get to being maximally entangled with $A$, measured by the square of the fidelity, $F^2$.

corresponds to a larger amount of decoherence. In the quantum case, $\mathrm{Dec}(A|E)$ can be computed using any semi-definite programming solver[29,30]. We remark that $\mathrm{Dec}(A|E)$ does itself not depend on the dimension of the system $A$. Furthermore, we note that $\mathrm{Dec}(A|E)$ does not depend on the particular physical realization of the system $A$, but merely the amount of information that it can hold. We point out that this entanglement-preservation picture is equivalent to the picture in which the quantum state of a single system decoheres[31] (see Fig. 4).

We hence see that in quantum mechanics, the relevant measure of decoherence is simply $\mathrm{Dec}(A|E)$ (see Fig. 5 for some examples). How can we estimate it in an experiment? Our goal in deriving this estimate will be to rely on concepts that we can later extend

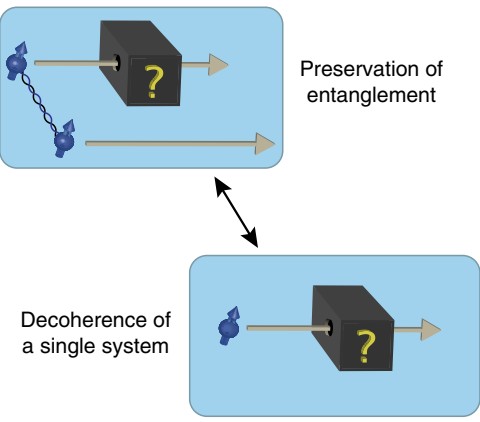

**Figure 4 | Equivalence of entanglement preservation and single system decoherence.** It is known[31] that decoherence on a single quantum system can be understood fully as the process' inability to preserve entanglement (for further background information see Supplementary Note 1). It is for this reason that our test for decoherence is fully general. In particular, it could also be applied to collapse models or any other form of decoherence. We emphasize gravitational decoherence, because here there is a pressing motivation for considering theories that modify or extend quantum mechanics.

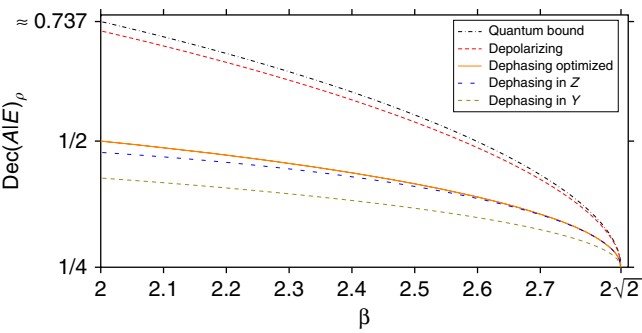

**Figure 5 | Comparison of the quantum bound with the actual values of $\mathrm{Dec}(A|E)_\rho$ for some example channels and measurements.** The black dash-dotted line on top shows the quantum bound, that is, the maximal value of $\mathrm{Dec}(A|E)_\rho$ that is compatible with a measured CHSH value $\beta$ in quantum theory. The other four plots are parametric plots: The parameter that is varied is the noise parameter of the channel (see Supplementary Notes 2 and 5). For each noise parameter, the value of $\mathrm{Dec}(A|E)_\rho$ of the resulting state is calculated, as well as the CHSH value $\beta$ that one would measure for this state using the standard measurements in the $X$-$Z$-plane that would be optimal for an EPR pair. This measurement happens to be optimal for the resulting state for the depolarizing channel, but not for the dephasing channels. The orange solid line also shows such a parametric plot for the dephasing channel, but for that line, the CHSH value $\beta$ is not calculated for the standard measurement for the EPR pair but for the measurement that is optimal for the actual resulting state[50]. The resulting curve is independent of the dephasing direction.

beyond the realm of quantum theory, deriving a universally valid test. It is clear that to estimate $\mathrm{Dec}(A|E)$ we need to make a statement about the entanglement between $A$ and $E$—yet $E$ is inaccessible to our experiment. A property of quantum mechanics known as the monogamy of entanglement[32] nevertheless allows such an estimate: if $\rho_{AB}$ is highly entangled, then $\rho_{AE}$ is necessarily far from highly entangled. Since low entanglement in $\rho_{AE}$ means that $\mathrm{Dec}(A|E)$ is low, a test that is able to detect entanglement between $A$ and $B$ should help us bound $\mathrm{Dec}(A|E)$ from above.

**Beyond quantum mechanics**. The real challenge is to show that the conclusions of our test remain valid even outside of quantum mechanics. Since we want to make as few assumptions as possible, we consider the most general probabilistic theory, in which we are only given a set of possible states $\Omega$ and measurements on these states. Every measurement is thereby a collection $M = \{e_a\}_a$ of effects $e_a{:}\Omega \to [0, 1]$ satisfying $e_a(\omega) \geq 0$ and $\sum_a e_a(\omega) = 1$ for all $\omega \in \Omega$. The label $a$ corresponds to a measurement outcome 'a'. The notion of separated systems $A$, $B$ and $E$ is in general difficult to define uniquely. We thus again make the most minimal assumption possible in which we identify 'systems' $A$, $B$ and $E$ with sets of measurements that can be performed. In a nutshell, we make the following assumptions: there is a notion of states and measurements, we can observe measurement outcomes that occur with some probability, we identify subsystems by sets of possible measurements, and the no-signalling principle holds (see Supplementary Notes 3 and 4 for details).

The first obstacle consists of defining a general notion of decoherence. We saw that quantumly decoherence can be quantified by how well correlations between $A$ and $A'$ are preserved, and this can be measured by how well the decoherence process preserves the maximally correlated (that is, entangled) state. Indeed, we can also quantify classical noise in terms of how well it preserves correlations, where the maximally correlated state takes on the form $(1/d_A) \sum_a |a\rangle\langle a|_A \otimes |a\rangle\langle a|_{A'}$ for some classical symbols $a$. We hence start by defining the set of maximally correlated states, by observing a crucial and indeed defining property of the maximally correlated state in quantum mechanics. Concretely, $A$ and $A'$ are maximally entangled if and only if for any von Neumann measurement on $A$, there exists a corresponding measurement on $A'$ giving the same outcome. Again, the same is also true classically but made trivial by the fact that there is only one measurement. In analogy, we thus define the set of maximally correlated states as

$$\Psi_{AA'} = \left\{ \Phi \in \Omega_{AA'} \,\middle|\, \forall M^A = \{e_a^A\}_a \exists M^B = \{e_a^B\}_a \text{ such that } \sum_a e_a^A e_a^B(\Phi) = 1 \right\} \tag{5}$$

This set coincides with the set of maximally entangled states in quantum mechanics, where $A'$ can potentially contain an additional component $\sigma_{A_2'}$ in $\Phi_{AA_1'} \otimes \sigma_{A_2'}$ which is irrelevant to our discussion. We thus define

$$\mathrm{Dec}(A|E)_\omega = \sup_{\mathcal{R}_{E \to A'}} \sup_{\Phi_{AA'} \in \Psi_{AA'}} F^2(\Phi_{AA'}, \mathcal{R}_{E \to A'}(\omega_{AE})) \tag{6}$$

where $\omega_{AE}$ is the state shared between $A$ and $E$ according to the general physical theory. The fidelity between two states $\omega_1$ and $\omega_2$ is thereby defined in full analogy to the quantum case[33] as

$$F(\omega_1, \omega_2) = \inf_M F(M(\omega_1), M(\omega_2)) \tag{7}$$

where the minimization is taken over all possible measurements $M$, and $M(\omega)$ denotes the probability distribution over the measurement outcomes of $M$. Here, the fidelity $F(M(\omega_1), M(\omega_2))$ can be written as[33]

$$F(M(\omega_1), M(\omega_2)) = \sum_i \sqrt{e_i(\omega_1)} \sqrt{e_i(\omega_2)} \tag{8}$$

where the sum ranges over all effects $e_i$ of the measurement $M$ (see Supplementary Note 3 for further details). That is, the fidelity can be expressed as the minimum fidelity between probability distributions of classical measurement outcomes. We will not need to make $\mathcal{R}_{E'}$ explicit in order to bound $\mathrm{Dec}(A|E)$. Equation (6) gives us the familiar quantity within quantum mechanics, but provides us with a very intuitive way to

quantify decoherence in any physical theory that admits maximally correlated states. We emphasize that with our general techniques the latter demand could be weakened to allow all theories, even those which only have (weak) approximations of maximally correlated states.

The second challenge is to prove that our test actually provides a bound on $\mathrm{Dec}(A|E)_\omega$. Note that without quantum mechanics to guide us, all that we could reasonably establish by performing measurements on $A$ and $B$ are the probabilities of outcomes $a$ and $b$ given measurement settings $x$ and $y$. That is, the probability

$$\mathrm{Pr}[a, b|x, y]_\omega = e_a^A e_b^B(\omega_{AB}) \tag{9}$$

where $e_a^A \in M_x^A$ and $e_b^B \in M_y^B$. Yet, given the system $E$ is entirely inaccessible to us we have no hope of measuring $\mathrm{Pr}[a, b, c|x, y, z]_\omega$ directly, where $z$ denotes a measurement setting on $E$ with outcome $c$. Nevertheless, similar to quantum entanglement, it is known that non-signalling distributions are again monogamous[34]—and it is this fact that allows us to draw conclusions about $E$ by measuring only $A$ and $B$. We will therefore make a non-trivial assumption about the physical theory, namely that no-signalling holds between $A$, $B$ and $E$. We emphasize that weaker constraints on the amount of signalling could also lead to a bound—but we are not aware of any other concrete example to consider. Mathematically, no-signalling means that the marginal distributions obey

$$\forall a, x, y, y', z, z' : \mathrm{Pr}[a|x, y, z]_\omega = \mathrm{Pr}[a|x, y', z']_\omega \tag{10}$$

that is, the choice of measurement settings $y$, $y'$ and $z$, $z'$ does not influence the probability distribution over the outcomes $a$. A set of distributions is non-signalling if such conditions hold for all marginal distributions.

## Discussion

What have we actually learned when performing such an experiment? We first observe that the measured $\beta$ always gives an upper bound on the amount of decoherence observed—for any non-signalling theory. This means that even if quantum mechanics would indeed need to be modified we can still draw conclusions from the data we obtain. As such, the observations made in such an experiment establish a fundamental limit on decoherence no matter what the theory might actually look like in detail. It is clear, however, that the bound thus obtained is much weaker than if we had assumed quantum mechanics. No-signalling is but one of many principles obeyed by quantum mechanics, and these other features put stronger bounds on the values that $\mathrm{Dec}(A|E)$ can take. Our motivation for considering theories which are only constrained by no-signalling is to demonstrate even such weak demands still allow us to draw meaningful conclusions from such an experiment. One can easily adapt our approach by introducing further constraints on the probabilities $\mathrm{Pr}[a, b, c|x, y, z]$—but not all of quantum mechanics—in order to get stronger bounds. Also in a fully quantum mechanical world, our approach yields a bound (see Fig. 6). If we assume quantum mechanics, we may of course also try and perform process tomography in order to determine the decoherence process, and indeed any experiment should try and perform such a tomographic analysis whenever possible. The appeal of our approach is rather that we can draw conclusions from the experimental data while making only very minimal assumptions about the underlying physical theory.

One may wonder why we only upper bound $\mathrm{Dec}(A|E)$. Note that from our experimental statistics we can only make statements about the overall decoherence observed in the experiment, namely the gravitational decoherence (if it exists) as well as any other decoherence introduced due to experimental imperfections. Finding that the Bell violation is low (and thus maybe $\mathrm{Dec}(A|E)$ might be large) can thus not be attributed conclusively to the

gravitational decoherence process, making a lower bound on Dec($A|E$) meaningless if our desire is to make statements about a particular decoherence process such as gravity.

Second, we observe that our approach can rule out models of gravitational decoherence but not verify a particular one. It is important to note that a model for gravitational decoherence does not stand on its own, but is always part of a theory on what states, evolutions and measurements behave like. Given such a physical theory and a model for gravitational decoherence, we know enough to compute Dec($A|E$), such as for example in equations (15–17). In addition, we can compute an upper bound $f_{\text{theory}}(\beta)$ on Dec($A|E$) specific to that theory, which may give a much stronger bound than no-signalling alone. Indeed, we see from Fig. 6 that this is the case for quantum mechanics. Given the calculated Dec($A|E$) and the experimentally observed value for $f_{\text{theory}}(\beta)$, we can then compare: If Dec($A|E$) > $f_{\text{theory}}(\beta)$, then the model (or indeed theory) we assumed must be wrong. However, if Dec($A|E$) ≤ $f_{\text{theory}}(\beta)$, then we know that the model and theory would be consistent without experimental observations.

Note that while our framework allows for theories with super-quantum correlations (that is, with $\beta > 2\sqrt{2}$ (ref. 35)), it is

also perfectly valid in the regime where $\beta \leq 2\sqrt{2}$. The bound shown in Fig. 6 is non-trivial for all $\beta > 2$, and therefore conclusions can be drawn for all such $\beta$. Hence, the various arguments brought forward in the literature for why super-quantum correlations should not be observed[36–42] do not contradict our work. The numeric value of the red bound in Fig. 6 may seem weak. However, recall from above that this is a bound for the most general class of theories that can be considered in our framework, while additional assumptions about the theory in question increase the strength of the bound.

Our approach thus provides a guiding light in the search for gravitational decoherence models. It is very general, and could in principle be used in conjunction with other proposed experimental setups and decoherence models. In particular, it could also be used to probe decoherence models conjectured to arise from decoherence affecting macroscopic objects, where there exist proposals to bring such objects into superposition[3]. Clearly, however, probing such models using entanglement is extremely challenging.

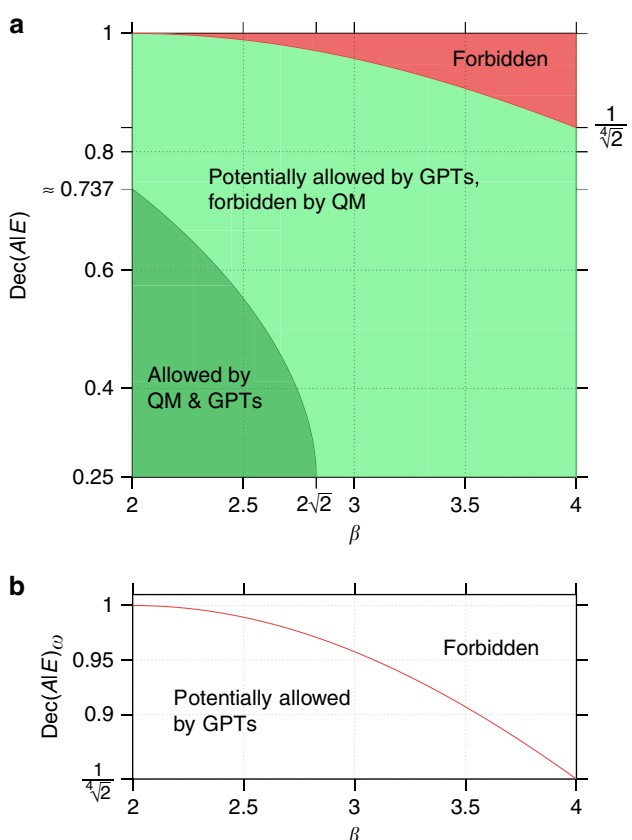

**Figure 6 | Allowed values of the decoherence quantity for measured CHSH values.** (**a**) shows what values of the decoherence quantity are compatible with some measured CHSH value $\beta$, assuming either quantum theory or any other probabilistic theory. The dark green region consists of all points ($\beta$, Dec($A|E$)$_\rho$) for which there exists a quantum state $\rho_{AB}$ and two pairs ($A_0$, $A_1$) and ($B_0$, $B_1$) of observables with the according values, that is, the bound is tight. The red region shows pairs ($\beta$, Dec($A|E$)$_\omega$) that cannot be realized in any non-signalling probabilistic theory. The curve between the light green area and the red area is a bound on Dec($A|E$)$_\omega$ which is valid for all non-signalling generalized probabilistic theories (GPTs). (**b**) shows a zoomed-in plot of the border line between the forbidden region and the region which is potentially allowed by GPTs. In a world constrained only by no-signalling, $\beta = 4$ is possible[51–54].

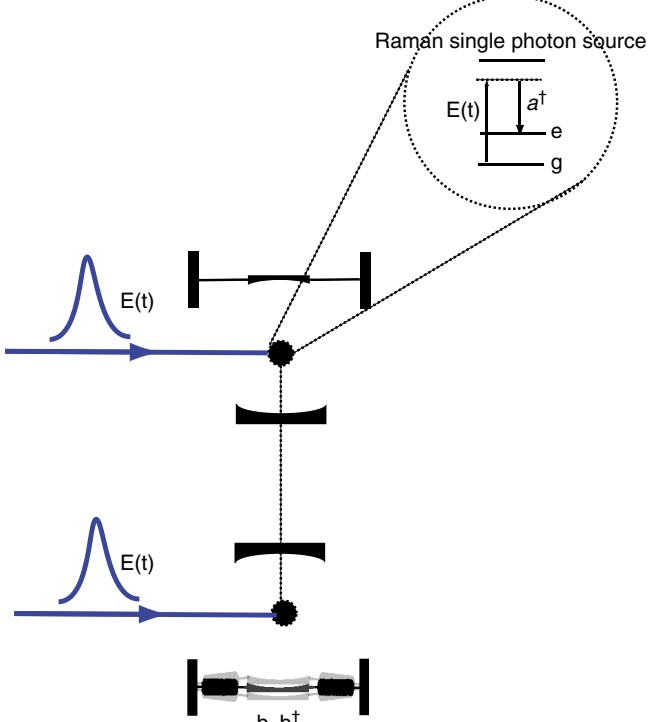

**Figure 7 | Probing an optomechanical system.** Our goal is to create entanglement between two optomechanical cavities. One cavity thereby has a movable mirror that introduces gravitational decoherence. Two cavities each contain a Raman single photon source controlled by an external laser 'write field' $E(t)$[55]. This write-field is used to map excitations in the atomic sources to single photon excitations in the cavities. The top cavity has fixed end mirrors while the bottom cavity has one mirror that is harmonically bound along the cavity axis and can move in response to the radiation pressure force of light in the cavity. The Raman sources are first prepared in an entangled state. This setup is a modification of the one proposed by Bouwmeester[2] in which an itinerant single photon pulse is injected into a cavity rather than created intra-cavity as here. Our modification avoids the problem that the time over which the photons interact with the mechanical element is stochastic and determined by the random times at which the photons enter and exit the cavity through an end mirror. In the new scheme, the cavities are assumed to have almost perfect mirrors—very narrow line width[56] (see Supplementary Note 6 for details).

It is a very interesting open question to improve our analysis and to apply it to other physical theories that are more constrained than by no-signalling, but yet do not quite yield quantum mechanics. Candidates for this may come from the study of generalized probabilistic theories where the authors (e.g., refs 43–48) introduced further constraints in order to recover quantum mechanics, but also from suggested ways to modify the Schrödinger equation in order to account for non-quantum mechanical noise. Since our approach could also be applied to higher dimensional systems, and other Bell inequalities, it is a very interesting open question whether other Bell inequalities could be used to obtain stronger bounds on $\text{Dec}(A|E)$ from the resulting experimental observations.

## Methods

**In quantum mechanics.** Figure 2 illustrates the general experimental procedure. As an easy warm-up, let us first again consider what happens in quantum mechanics. For now, we assume that the measurement devices have no memory. That is, the experiment behaves the same in each round, independent on the previous measurements. It is relatively straightforward to obtain an upper bound on $\text{Dec}(A|E)$ by extending techniques from quantum key distribution[49]. In essence, we maximize $\text{Dec}(A|E)$ over all states that are consistent with the observed CHSH correlator $\beta$ (see Fig. 2). This maximization problem is simplified by the inherent symmetries of the CHSH inequality, allowing us to reduce this optimization problem to consider only states that are diagonal in the Bell basis. We proceed to establish properties of min and max entropies for Bell diagonal states, leading to an upper bound. Concretely, we show in Supplementary Note 2 that

$$\text{Dec}(A|E) \leq h(\beta) \tag{11}$$

where $h(\beta)$ is an easy optimization problem that can be solved using Lagrange multipliers. We have chosen not to weaken this bound by an analytical bound that is strictly larger, as it is indeed easily evaluated (see Fig. 6). If the devices are allowed memory, then a variant of this test and some more sophisticated techniques from quantum key distribution can nevertheless be shown to give a bound.

**Beyond quantum mechanics.** Let us first give a very loose intuition why performing a Bell experiment on A and B may allow us to bound $\text{Dec}(A|E)_\omega$. It is well known[34] that non-signalling correlations are also monogamous. That is, if we observe a violation of the CHSH inequality as captured by the measured parameter $\beta$, then we know that the violation between A and E and also between E and B must be low. Note that the expectation values $\text{Tr}[\rho_{AB}(A_x \otimes B_y)]$ in terms of quantum

observables $A_x$ and $B_y$ can be expressed in terms of probabilities as

$$\text{Tr}\big[\rho_{AB}\big(A_x \otimes B_y\big)\big] = \sum_{a \in \{\pm 1\}} \Pr[a, a|x, y]_\omega - \Pr[a, -a|x, y]_\omega \tag{12}$$

where we have again used $\omega_{AB}$ in place of $\rho_{AB}$ to remind ourselves that we may be outside of QM. Let us now assume by contradiction that the state $\omega_{AE}$ shared between A and E would be close to maximally correlated. Then by definition of the maximally correlated state, for every measurement on A, there exists some measurement on E which yields the same outcome with high probability. Hence, if $\omega_{AE}$ would be close to maximally correlated, then we would expect that E and B can achieve a similar CHSH violation as A and B—because E can make measurements that reproduce the same correlations that A can achieve with B. Yet, we know that this cannot be since CHSH correlations are monogamous. Note that a map $\mathcal{R}$ (as in Fig. 3), followed by a measurement in fact constitutes another measurement. Hence, considering all possible measurements that Eve can perform, we cover all such possible maps $\mathcal{R}$ that Eve might want to apply.

While we do not follow the exact steps suggested by this intuition, we employ a technique in Supplementary Note 3 that has also been used for studying monogamy of CHSH correlations[34]. Specifically, we use linear programming as a technique to obtain bounds. We thereby first relate the fidelity to the statistical distance, which is a linear functional. We are then able to optimize this linear functional over probability distributions $\Pr[a, b, c|x, y, z]_\omega$ satisfying linear constraints. The first such constraint is given by the fact that we consider only non-signalling distributions. The second is the fact that the marginal distribution $\Pr[a, b|x, y]_\omega$ leads to the observed Bell violation $\beta$. The last one stems from the fact that maximal correlations can also be expressed using a linear constraint. Solving this linear program for an observed violation $\beta$ leads to Fig. 6.

**Optomechanical experiment.** To gain insights into the significance of gravitational decoherence, we examine Diosi's theory of gravitational decoherence[6] as an example. This is equivalent to the decoherence model introduced in Kafri et al.[10]. We show in Supplementary Note 6 how $\text{Dec}(A|E)$ can be evaluated for many other decoherence processes, opening the door for applying our method to many other possible experiments. Diosi's model can be applied to an optomechanical cavity in which one mirror is free to move in a harmonic potential with frequency $\omega_m$ as in Fig. 7. The master equation for a massive particle moving in a harmonic potential, including gravitational decoherence is

$$\frac{d\rho}{dt} = -i\omega_m\big[b^\dagger b, \rho\big] - \Lambda\big[b + b^\dagger, \big[b + b^\dagger, \rho\big]\big] \tag{13}$$

where

$$b = \sqrt{\frac{m\omega_m}{2\hbar}}\hat{x} + i\frac{1}{\sqrt{2\hbar m\omega_m}}\hat{p} \tag{14}$$

with $\hat{x}, \hat{p}$ the usual canonical position and momentum operators for the moving mirror. We have that

$$\Lambda = \Lambda_{\text{grav}} + \Lambda_{\text{heat}} \tag{15}$$

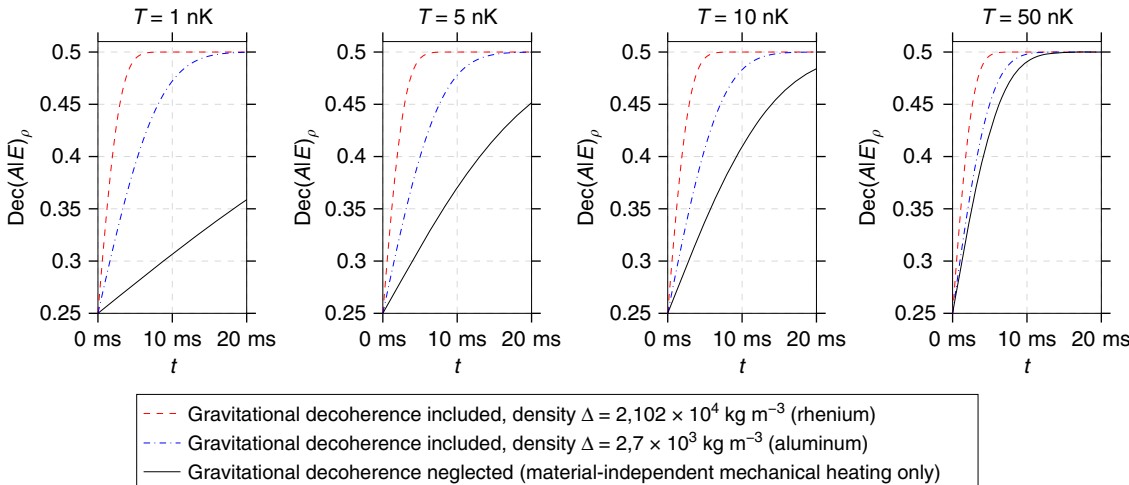

**Figure 8 | Predicted values of the decoherence quantity in the optomechanical experiment.** This figure shows the predicted values of $\text{Dec}(A|E)_\rho$ as a function of the running time of the optomechanical experiment for different temperatures and for different materials of the mechanical element as calculated in the proposed model for gravitational decoherence. In addition, $\text{Dec}(A|E)_\rho$ is plotted for the case where gravitational decoherence is not taken into account. When the gap between the predicted values with and without gravitational decoherence is large enough, the decoherence estimation formalism allows for a test that potentially falsifies the proposed model for gravitational decoherence. The calculations have been made for the example experimental parameters $g_0 = 1\,\text{s}^{-1}$, $\omega_m = 1\,\text{s}^{-1}$ and $\gamma_m = 10^{-10}\,\text{s}^{-1}$.

where the gravitational decoherence rate $\Lambda_{grav}$ is given by

$$\Lambda_{grav} = \frac{2\pi G \Delta}{3} \frac{1}{\omega_m} \qquad (16)$$

with $G$ the Newton gravitational constant and $\Delta$ the density of the moving mirror. As one might expect $\Lambda_{grav}$ is quite small, of the order of $10^{-8} \, s^{-1}$ for suspended mirrors with $\omega_m \sim 1$. The term

$$\Lambda_{heat} = \frac{k_B T}{\hbar Q} \qquad (17)$$

with $Q = \omega/\gamma_m$ corresponds to mechanical heating. To see the effect of the gravitational term stand out next to the mechanical heating we thus need to make the temperature $T$ low. A calculation shows that this model leads to a dephasing channel $\Gamma(\rho) = p\rho + (1-p)Z\rho Z^\dagger$ where $p$ is a function of the density $\Delta$, and the other parameters. In Supplementary Note 6, we show that for this model

$$\text{Dec}(A|E)_\rho = \frac{1}{4}\left(1 + \sqrt{1 - \exp\left(-4\left(1 + 2\left(\frac{4\pi G}{3}\frac{1}{\gamma_m \omega_m}\Delta + \frac{2k_B}{\hbar}\frac{1}{\omega_m}T\right)\right)\frac{g_0^2}{\omega_m^2}\sin^2\left(\frac{\omega_m t}{2}\right)\right)}\right) \qquad (18)$$

where $G$ is the Newton gravitational constant, $k_B$ is the Boltzmann constant, and $\hbar$ the Planck constant (see Fig. 8 for the other parameters).

**Code availability.** The source code of the semidefinite program and the linear program used to derive the plots in Fig. 6 are available from the authors on request.

**Data availability.** Data sharing not applicable to this article as no data sets were generated or analysed during the current study.

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

## Acknowledgements

We thank Markus P. Müller, Matthew Pusey, Tobias Fritz, Gary Steele, Jonas Helsen and Thinh Le Phuc for insightful discussions. C.P., J.K., M.T., A.M., R.S. and S.W. were supported by MOE Tier 3A grant 'Randomness from quantum processes', NRF CRP 'Space-based QKD'. S.W. was also supported by STW, Netherlands, an NWO VIDI, and an ERC Starting Grant. N.M. and G.M. were supported by ARC Centre of Excellence for Engineered Quantum Systems, CE110001013.

## Author contributions

S.W. devised the project, the main conceptual ideas and proof outline. C.P. worked out almost all of the technical details, and performed the numerical calculations for the

suggested experiment. J.K. worked out the bound for quantum mechanics, with help from M.T. and A.M. R.S. verified the numerical results of the linear program by an independent implementation. N.M. and G.M. proposed the optomechanical experiment in discussions with S.W. C.P., J.K., G.M. and S.W. wrote the manuscript.

## Additional information

**Competing financial interests:** The authors declare no competing financial interests.

