## [Peer Review File · Nature Communications]

Reviewers' comments:

Reviewer #1 (Remarks to the Author):

Review NCOMMS-16-01995 Pfister

The aim of this paper is to suggest a way for experiments to address the question of whether quantum decoherence, or even "post-quantum" decoherence, could be due to gravity. The main idea, as I understand it, is the following: The decoherence of a nonlocal, i.e. entangled, state would change it from a state that violates Bell's inequality, and has value as for various quantum-information tasks, to one that does not. Since there are tests for the degree of entanglement of quantum states, it might be possible to measure the effect of decoherence (due to whatever mechanism) by finding spontaneous degradation of this nonlocal resource. The most interesting and novel part of the paper is the derivation of bounds on decoherence that could arise from non-signalling correlations that are stronger than nonlocal quantum correlations. I could recommend publication of this work but I have the following criticisms.

My initial criticism of this paper is that although evidently much work has gone into it, it does come across as a paper meant to be read, because the reader is not given the tools to read it. For example, the sentence including Eq. (1) advertises $\text{Dec}(A|E)$ as a measure of coherence to be defined below. However, although $(A|E)$ appears in the next section as well in the form $H_{\min}(A|E)$, neither A nor E is ever defined. The reference to Fig. 2 does not help, since they are not defined there either, or in the caption. Also, the connection between $|\Psi_{ABE}\rangle$ and Fig. 2, to which it is referred, is anybody's guess. Is that why Fig. 2 is full of question marks? (And for those who read the Supplementary Information: E denotes Eve? Environment?)

I may be nitpicking but when we get to Eq. (3), which seems really essential to the development, there is a function F^2 that is nowhere defined, except in the Supplementary Information, which apparently is required reading. The reader is not even directed to the Supplementary Information.

I should also raise material questions unrelated to this one. First, let's suppose that the proposed procedure does turn up evidence of decoherence; how can it be traced specifically to gravity? Could it not be due e.g. to the GRW-Pearle mechanism, which modifies quantum mechanics? And why is there no mention of the work of I. Pikovski, M. Zych, F. Costa and C. Brukner, (2015), <http://dx.doi.org/10.1038/nphys3366>, which does not involve a modification of quantum mechanics? Second, I would expect this paper to at least consider the claim in arXiv:1408.3125v1 to the effect that nonlocal correlations that are stronger than quantum correlations are unphysical (hence not worth testing).

In sum, I can certainly imagine that a revised version of this paper would be suitable for publication in a top physics journal. As far as I know, papers in Nat. Commun. are not intended for a broad audience but rather for specialists in various fields, in which case I could imagine that it would be suitable for Nat. Commun. as well.

Reviewer #2 (Remarks to the Author):

Motivated by the searches for quantum gravity signatures, the authors quantify decoherence both in quantum mechanics (QM) and generalized probability theories (GPT)

In QM they use min- entropy, its expression through fidelity, and the entanglement/CHSH inequality violation tests.

In GPB they first provide reasonable definitions of coherence, coherence and fidelity, using only the minimal set of assumptions [the no-signalling is probably the strongest one], and express the quantities of interests in terms of classical data -- the measurement results. QM then naturally appears as a special case.

Already the QM part of the paper is, to my knowledge, new. Discussions of the optomechanical experiments are done in this framework and provide a new angle to consider some particular models.

Here I have to admit that I had a great difficulty in following the manuscript, not only in many of the technical details but even the logic. Some specific comments about the presentation are given below.

I was not able to understand the apparently main claim of the paper: a possibility to extract model-independent quantitative bounds on decoherence, making only the basic assumptions that were mentioned above. Already in the QM case [Methods] the quantitative conclusions are obtained only when a specific decoherence model is used; different models under the same circumstances will provide different gravitational decoherence rate (Eqs 13, 14).

Hence I simply do not see how meaningful conclusions can be drawn in a GPT case without being sufficiently specific on the types of maps \mathcal{R} (as in Eq 5).

Presentation:

Both structure and notation are confusing. After reading the whole manuscript ((and parts of the SI) the reason for first discussing the QM case, and then the GPT case become perfectly clear. However, a brief opening statement of intentions after Eq 1: to abstract the essential features of QM case and then generalize them for the GPT would be very welcome.

The naming and renaming of the subsystems is very confusing [in a single paragraph before Eq 2 A' becomes both B and E1, etc), while fidelity (Eq 3) is not only undefined, but is not even named. If the reader is not familiar with Hmin, absence of brackets in Eq 2 may be misleading. In general, the authors should carefully distinguish (and refer to) purification of a state, initial and final states/spaces of the system, etc. [For example, this is not stated in several key parts of Sec 1.2, like Eq 1.9] While the generalization (different dimensions for S and B) are not used in the manuscript and can be skipped, Fig 2 of the SI would be actually welcome in the manuscript.

In my view applying Hmin to the gravitational decoherence problem in the framework of QM, if done properly, would be a reason to seriously consider publication. However, I do not see how the main claims of the manuscript are substantiated, and I cannot recommend publication in its present form.

Reviewer #3 (Remarks to the Author):

I find this paper rather interesting.

And while I could not find time to redo all calculations my impression is that they are correct. Still it does not look to me as NatureComm material.

I would expect a more significant sharper "take home message" from a NatureComm publication. It is interesting that one can do properly data analysis applicable to any physical theory where the no-

signaling principle hold. But how much will that change research on quantum-gravity-induced decoherence?

And is it really necessary?

Physics has found ways to discover new theoretical paradigms even working within the previous theoretical paradigm. This is actually what typically happens. Quantum Mechanics itself was discovered in that way, and even after studying this manuscript I still expect that it will be in that way that Quantum Mechanics will be replaced.

Response to Reviewer #1

We thank Reviewer #1 for his helpful comments and we are delighted to see that he/she is positive about our work. We have refined our manuscript in several ways, and respond to the referee below.

The aim of this paper is to suggest a way for experiments to address the question of whether quantum decoherence, or even "post-quantum" decoherence, could be due to gravity. The main idea, as I understand it, is the following: The decoherence of a nonlocal, i.e. entangled, state would change it from a state that violates Bell's inequality, and has value as for various quantum-information tasks, to one that does not. Since there are tests for the degree of entanglement of quantum states, it might be possible to measure the effect of decoherence (due to whatever mechanism) by finding spontaneous degradation of this nonlocal resource. The most interesting and novel part of the paper is the derivation of bounds on decoherence that could arise from non-signalling correlations that are stronger than nonlocal quantum correlations. I could recommend publication of this work but I have the following criticisms.

In sum, I can certainly imagine that a revised version of this paper would be suitable for publication in a top physics journal. As far as I know, papers in Nat. Commun. are not intended for a broad audience but rather for specialists in various fields, in which case I could imagine that it would be suitable for Nat. Commun. as well.

We thank the referee for his support.

The referee writes further:

My initial criticism of this paper is that although evidently much work has gone into it, it does come across as a paper meant to be read, because the reader is not given the tools to read it. For example, the sentence including Eq. (1) advertises $\text{Dec}(A|E)$ as a measure of coherence to be defined below. However, although $\text{Dec}(A|E)$ appears in the next section as well in the form $H_{\min}(A|E)$, neither A nor E is ever defined. The reference to Fig. 2 does not help, since they are not defined there either, or in the caption. Also, the connection between $|\Psi_{ABE}\rangle$ and Fig. 2, to which it is referred, is anybody's guess. Is that why Fig. 2 is full of question marks? (And for those who read the Supplementary Information: E denotes Eve? Environment?)

We agree that we could be more clear, and we thank the referee for raising this point. It seems that in the previous version of our manuscript, we were not sufficiently clear about the system labels in the introductory part of the document. We fixed this by changing the sentence from

[...], where $\text{Dec}(A|E)$ is the amount of decoherence defined below.

to

[...], where $\text{Dec}(A|E)$ is the amount of decoherence of a system A with respect to its environment E (we will define this below).

Moreover, we changed Fig. 2: we added system labels in the figure, and placed the symbol of the state Ψ_{ABE} next to the relevant systems. In order to match the figure more with the caption, we also included the input system E_{in} of the environment into the picture. This also makes it easier to see the connection to Fig. 3 that we added in the revised manuscript according to the advice of Reviewer #2 (see below).

To keep the notation consistent, we have also changed the label S to A' in Section I.1 of the Supplementary Information, so that the labeling conventions in the two documents coincide and added a footnote that explains this. It reads:

Here we denote the system by A' rather than A because we reserve the label for another system (see Figure 2 and 3). This way, the labeling convention of the main article coincides with the labeling in the Supplementary Information.

We also changed the first paragraph of Section 1.2 of the Supplementary Information, which now reads:

A lot can be learned about the channel if one takes an additional system into the picture. From now on, we consider the case where the input system A' is purified by a system A . Then, while system A' undergoes the channel evolution, system A remains unchanged. For example, one such case is the case where the system AA' is in a maximally entangled state. This leads to the situation shown in Figure 2. The output system is now a tripartite system ABE . In quantum information science, it is very popular to think of the systems as being controlled by parties with intentions and interests rather than just being dead physical objects. We will follow this spirit and from now on use the language of a game and speak of parties Alice, Bob and Eve, that we think of as agents controlling the systems A , B and E . In quantum information science, it has been realized that important quantitative measures of the channel are functions of this tripartite state ρ_{ABE} .

We added a small figure there (Fig. 2) that helps following the labelings.

We are confident that with these changes, the system labelings are more clear now.

Next, the referee writes:

I may be nitpicking but when we get to Eq. (3), which seems really essential to the development, there is a function F^2 that is nowhere defined, except in the Supplementary Information, which apparently is required reading. The reader is not even directed to the Supplementary Information.

We agree we could have been more clear. To clarify that F denotes the fidelity, we added the following to the sentence where F occurs for the first time (Eqn. 3):

and where F denotes the fidelity [26]

$$F(\rho, \sigma) = \text{tr}(\sqrt{\sqrt{\rho} \sigma \sqrt{\rho}}). \text{ (Eqn. 4)}$$

Reference [26] is a textbook on quantum information theory that explains the fidelity in great detail. In addition, we added the following below Eqn. 8 (formerly Eqn. 7):

Here, the fidelity $F(M(\omega_1), M(\omega_2))$ can be written as [26]

$$F(M(\omega_1), M(\omega_2)) = \sum_i \sqrt{e_i(\omega_1)} \sqrt{e_i(\omega_2)},$$

where the sum ranges over all effects e_i of the measurement M (see the Supplementary Information for further details).

Reviewer #2 also raises the following question regarding our manuscript:

I should also raise material questions unrelated to this one. First, let's suppose that the proposed procedure does turn up evidence of decoherence; how can it be traced specifically to gravity? Could it not be due e.g. to the GRW-Pearle mechanism, which modifies quantum mechanics?

We note that we quite clearly state that we only measure the total amount of decoherence, and we can indeed not say anything about its origin. To make sure that the decoherence is due to gravity and not to anything else, it is important to eliminate other forms of decoherence as much as possible. It is for this reason that in our optomechanical example we need to achieve such low temperatures for example.

We also remark that we discuss this point in detail in the discussion. Specifically this is also the reason why we derive only an upper bound on the amount of decoherence – that is, the decoherence is never larger than some measured number. This bound applies universally and does allow us to draw conclusions about gravitational decoherence (or other forms of decoherence): if the total amount of decoherence does not exceed a certain number than neither do the contributions from gravity. It does not make much sense in an experiment to conclude that the decoherence is at least something: there might be decoherence quite simply due to a faulty experimental apparatus and we cannot draw definite conclusions about gravity.

We quote from our explanation in the Discussion section:

“One may wonder why we only upper bound $\text{Dec}(A|E)$. Note that from our experimental statistics we can only make statements about the overall decoherence observed in the experiment, namely the gravitational decoherence (if it exists) as well as any other decoherence introduced due to experimental imperfections. Finding that the Bell violation is low (and thus maybe $\text{Dec}(A|E)$ might be large) can thus not be attributed conclusively to the gravitational

decoherence process, making a lower bound on $Dec(A|E)$ meaningless if our desire is to make statements about a particular decoherence process such as gravity.”

We would like to remark that our bound does allow us to rule out specific models of decoherence (if they turn out to be incorrect). We quote from the Discussion:

“Second, we observe that our approach can rule out models of gravitational decoherence but not verify a particular one. It is important to note that a model for gravitational decoherence does not stand on its own, but is always part of a theory on what states, evolutions and measurements behave like. Given such a physical theory and a model for gravitational decoherence, we know enough to compute $Dec(A|E)$. In addition, we can compute an upper bound $f_{theory}(\beta)$ on $Dec(A|E)$ specific to that theory, which may give a much stronger bound than no-signalling alone. Indeed, we see from Figure 5 that this is the case for quantum mechanics. Given the calculated $Dec(A|E)$ and the experimentally observed value for $f_{theory}(\beta)$, we can then compare: If $Dec(A|E) > f_{theory}(\beta)$, then the model (or indeed the theory) we assumed must be wrong. However, if $Dec(A|E) \leq f_{theory}(\beta)$, then we know that the model and theory would be consistent with our experimental observations.”

Next, the referee writes:

And why is there no mention of the work of I. Pikovski, M. Zych, F. Costa and C. Brukner, (2015), <http://dx.doi.org/10.1038/nphys3366>, which does not involve a modification of quantum mechanics?

We apologize for this oversight and have added the reference.

Second, I would expect this paper to at least consider the claim in arXiv:1408.3125v1 to the effect that nonlocal correlations that are stronger than quantum correlations are unphysical (hence not worth testing).

We kindly note that we make no statements about what theories are valid or not. In fact, our work precisely aims to find a universal bound on decoherence without having to specify the exact model.

Most importantly, we would kindly like to point out that even if β is constrained to values not greater than $2\sqrt{2}$ – hence ruling out stronger than quantum correlations - our framework applies and provides a potentially very useful tool in the search for a theory that explains gravitational decoherence. Our approach is very general and we could even consider models which are no signaling, constraint correlations to be the same as in quantum mechanics, but add all kinds of other constraints coming from the theory in question.

To emphasize this, we added the following paragraph to the “Discussion” section:

Note that while our framework allows for theories with super-quantum correlations (that is, with $\beta > 2\sqrt{2}$ [40]), it is also perfectly valid in the regime where $\beta < 2\sqrt{2}$. The bound shown in Fig. 6 is non-trivial for all $\beta > 2$, and therefore conclusions can be drawn for all such β . Hence, the various arguments brought forward in the literature for why super-quantum correlations should not be observed [41–47] do not contradict our work. The numeric value of the red bound in Fig. 6 may seem weak. However, recall from above that this is a bound for the most general class of theories that can be considered in our framework, while additional assumptions about the theory in question increase the strength of the bound.

We hope that these changes clarify the manuscript and that our response answers the questions raised by Reviewer #1.

Response to Reviewer #2

We thank Reviewer #2 for his helpful comments, and – while our paper is of course much more general – we are indeed pleased to see that he already likes just the quantum mechanical part, which we provided as a warmup. We have refined our manuscript in several ways, and respond to the referee below.

The referee writes:

Motivated by the searches for quantum gravity signatures, the authors quantify decoherence both in quantum mechanics (QM) and generalized probability theories (GPT)

In QM they use min- entropy, its expression through fidelity, and the entanglement/CHSH inequality violation tests.

In GPB they first provide reasonable definitions of coherence, coherence and fidelity, using only the minimal set of assumptions [the no-signalling is probably the strongest one], and express the quantities of interests in terms of classical data -- the measurement results. QM then naturally appears as a special case.

Already the QM part of the paper is, to my knowledge, new. Discussions of the optomical experiments are done in this framework and provide a new angle to consider some particular models.

Here I have to admit that I had a great difficulty in following the manuscript, not only in many of the technical details but even the logic. Some specific comments about the presentation are given below.

I was not able to understand the apparently main claim of the paper: a possibility to extract model-independent quantitative bounds on decoherence, making only the basic assumptions that were mentioned above. Already in the QM case [Methods] the quantitative conclusions are obtained only when a specific decoherence model is used; different models under the same circumstances will provide different gravitational decoherence rate (Eqs 13, 14).

We respectfully point out that the referee seems to be mixing the general bound and the calculation of Dec for an example. First, we derive a general upper bound on Dec – without having to know anything about the decoherence process. As we explain in the section “Beyond QM”, the general bound that we derive is valid for essentially any theory that obeys the no-signalling principle. The assumptions are summarized in the first paragraph of the “Beyond QM” sections, and are explained in great detail in Sections 3.1 and 3.2 of the Supplementary Information. This is a *general upper bound* with an *experimental input*. A sharper bound can be obtained if we also assume QM.

Eqns. 13 and 14, on the other hand, are concerned with testing the predictions of a particular decoherence model – and given the model we can calculate what Dec would be according to that model. This way, the theoretically predicted value can be compared with our experimental upper bound. If the predicted

value violates the bound, then the particular theory in question is ruled out. This is one strength of our framework: it can serve as a falsification tool for tested theories for gravitational decoherence.

We have already emphasized this at the beginning of the “Discussion” section, which we have now extended to put (13) and (14) in the right context:

“What have we actually learned when performing such an experiment? We first observe that the measured β always gives an upper bound on the amount of decoherence observed for any non-signalling theory. This means that even if quantum mechanics would indeed need to be modified we can still draw conclusions from the data we obtain. As such, the observations made in such an experiment establish a fundamental limit on decoherence no matter what the theory might actually look like in detail. [...] Our motivation for considering theories which are only constrained by no-signalling is to demonstrate even such weak demands still allow us to draw meaningful conclusions from such an experiment.”

A bit further in the same section, we write the following (note that equations (13) and (14) correspond to equations (15) and (16) in the new version; moreover, we also refer to equation (17), which was already there before, because it belongs with equations (15) and (16)) :

“Second, we observe that our approach can rule out models of gravitational decoherence but not verify a particular one. It is important to note that a model for gravitational decoherence does not stand on its own, but is always part of a theory on what states, evolutions and measurements behave like. Given such a physical theory and a model for gravitational decoherence, we know enough to compute $\text{Dec}(A|E)$, such as for example in equations (15) to (17). In addition, we can compute an upper bound $f_{\text{theory}}(\beta)$ on $\text{Dec}(A|E)$ specific to that theory, which may give a much stronger bound than no-signalling alone. Indeed, we see from Figure 5 that this is the case for quantum mechanics. Given the calculated $\text{Dec}(A|E)$ and the experimentally observed value for $f_{\text{theory}}(\beta)$, we can then compare: If $\text{Dec}(A|E) > f_{\text{theory}}(\beta)$, then the model (or indeed the theory) we assumed must be wrong. However, if $\text{Dec}(A|E) \leq f_{\text{theory}}(\beta)$, then we know that the model and theory would be consistent with our experimental observations.”

The next question of the referee concerns the maps $\text{cal}\{R\}$:

Hence I simply do not see how meaningful conclusions can be drawn in a GPT case without being sufficiently specific on the types of maps $\text{cal}\{R\}$ (as in Eq 5).

The maps $\text{cal}\{R\}$ do not need to be specified further: any such map leads to non-signalling statistics, and we prove our bounds for all such statistics. We show this in detail in the proof of Proposition 3.18 in the Supplementary Information. The fact that not knowing the types of maps is not a problem can also be seen as follows: a map, followed by a measurement, constitutes effectively another measurement. Since we are optimizing over all measurements, all such maps are covered in our optimization. We have also

added an additional sentence to the main file to highlight this and reference the supplementary material:

Note that a map R (as in Figure 3), followed by a measurement in fact constitutes another measurement. Hence, considering all possible measurements that Eve can perform, we cover all such possible maps R that Eve might want to apply.

In addition, Reviewer #2 makes very useful comments on the presentation of our result. He or she writes:

After reading the whole manuscript ((and parts of the SI) the reason for first discussing the QM case, and then the GPT case become perfectly clear. However, a brief opening statement of intentions after Eq 1: to abstract the essential features of QM case and then generalize them for the GPT would be very welcome.

We added a sentence to the beginning of the QM section. Now it reads:

As an easy warm-up, let us first focus on the concept of decoherence within quantum mechanics. This demonstrates some principles that we will generalize to a broad framework of theories in the following section.

Then, the referee writes:

The naming and renaming of the subsystems is very confusing [in a single paragraph before Eq 2 A' becomes both B and $E1$, etc), while fidelity (Eq 3) is not only undefined, but is not even named.

Reviewer #2 comments on the lack of clarity about our system labeling, just as Reviewer #1 did. Our reply on this largely coincides with the one for Reviewer #1, and we are confident that with the changes that we made (as outlined above), the labeling are much clearer now. The particular paragraph mentioned by Reviewer #2 should, with the help of Fig. 3, be much clearer now. The remark about the fidelity also coincides with the remark of Reviewer #1, to which we have replied and edited the manuscript accordingly.

Further, the referee remarks:

If the reader is not familiar with H_{min} , absence of brackets in Eq 2 may be misleading.

We agree with the referee. In order to avoid ambiguity, we added the brackets.

Next, the referee notes:

In general, the authors should carefully distinguish (and refer to) purification of a state, initial and final states/spaces of the system, etc. [For example, this is not stated in several key parts of Sec 1.2, like Eq 1.9]

We thank Reviewer #2 for his attentive reading of our manuscript. This is a very helpful remark, and we have modified Section 1.2 of the Supplementary Information accordingly.

Another useful remark by Reviewer #2 is the following:

While the generalization (different dimensions for S and B) are not used in the manuscript and can be skipped, Fig 2 of the SI would be actually welcome in the manuscript.

Again, we find that this is very useful input for the presentation of our work. We agree that this figure helps a lot in the main article. We created a new figure (Fig. 3), which is essentially the same figure as Fig. 2 of the appendix, with its style adjusted to the other figures of the main article. We are confident that this provides additional aid in understanding our work. Moreover, we find that this figure is very useful to clarify the labeling of the systems.

Response to Reviewer #3

We thank the reviewer for calling our paper rather interesting.

I find this paper rather interesting.

And while I could not find time to redo all calculations my impression is that they are correct.

Still it does not look to me as NatureComm material.

I would expect a more significant sharper "take home message" from a NatureComm publication. It is interesting that one can do properly data analysis applicable to any physical theory where the no-signaling principle hold. But how much will that change research on quantum-gravity-induced decoherence?

We believe that an estimate of decoherence which is universal – and does not require knowledge of the particular gravitational decoherence process – is precisely greatly helpful in guiding the search for the right model of such processes.

And is it really necessary? Physics has found ways to discover new theoretical paradigms even working within the previous theoretical paradigm. This is actually what typically happens. Quantum Mechanics itself was discovered in that way, and even after studying this manuscript I still expect that it will be in that way that Quantum Mechanics will be replaced.

To find the right model we urgently require actual experimental data, without which the right model cannot be decided. We believe that our work contributes significantly to these efforts by allowing us to draw universally valid conclusions from such experiments, which are valid for any model or even physical theory (even the no signaling condition could in principle be relaxed, or additional constraints introduced as we remark in the text – one only needs good motivation for doing so, which is why we focused merely on the no signaling case).

Given that there exists a rift between quantum mechanics and general relativity, we do not know what will be the right answer in the end. That is, we do not know whether we need completely new insights or can use ideas from quantum mechanics – and we are not aware that anyone knows. Such speculations, however, are not the objective of our article – our article merely aims to contribute to answering this question by providing a universal way to measure gravitational decoherence to guide the development of a unifying theory.

REVIEWERS' COMMENTS:

Reviewer #1 (Remarks to the Author):

I was pleasantly surprised by the improved clarity of the paper. There are nevertheless a few minor improvements I can suggest.

1. The caption to Fig. 1 ends with a reference to "theories that allow a small amount of signaling." I don't know what the authors mean or whether it could even make sense. Certainly in quantum mechanics, any signaling arising from correlations would inevitably violate relativity. The authors should either explain what they have in mind or leave this reference out.
2. I don't know what the notation $E=E_1 E_2$ just above Eq. (2) means.
3. The caption to Fig. 2 mentions E_{in} which is nowhere defined (as far as I can see). Better use the notation of the text. Also, the caption mentions "The channel's ability to preserve quantum information - that is, the amount of decoherence", which seems backwards. Doesn't the word "coherence" parallel "quantum information" better than "decoherence"?
4. Similarly, the caption to Fig. 4 states that "decoherence on a single quantum system can be understood fully by how well the decoherence process preserves entanglement", which is also backwards. Decoherence does not preserve entanglement, so I suggest "destroys" or a synonym instead of "preserves".

Reviewer #2 (Remarks to the Author):

The authors made a number of important changes in presentation, both better explaining the motivation and carefully explaining (or, at least, defining) the relevant quantities and subsystems.

My concerns about validity of the proposed upper bounds are resolved, and I have no further objections about the technical details and the presentation.

Regarding the significance of this work, after reading the authors response and reading of the improved version of the manuscript, I am convinced that it deals with an important question (how to deal with potential consequences of quantum gravity even if we might to make changes to quantum mechanics). The authors provide a very useful and inspiring insight on this issue, and I recommend to publish the manuscript in its present form.

Reviewer #3 (Remarks to the Author):

I find that the refereeing process is improving the manuscript, also thanks to recommendations by other referees. This is a good interesting paper that will surely influence rather strongly the community interested in such studies, but I remain skeptical that it will have appeal extending beyond that community. This broader appeal would have been achieved for example if the authors had provided compelling arguments essentially explaining why for this next revolution it should be so important to have the most appropriate tools for interpretation of data.

I do not find such a compelling case in the manuscript or in the reply, and I would still take as reference the fact that for all previous major "physics revolutions" we (the physicists of those times) did just fine employing rather inadequate tools of data analysis (like using classical-mechanics-optimized data analysis for many analyses that led to the discovery of quantum mechanics).

Response to Reviewer #1

We thank Reviewer #1 for mentioning the improved clarity of our manuscript. In addition, he / she provided very useful comments for further improvements of our work by pointing out four remaining issues. In the following, we cite the reviewer's comments and explain how we addressed them.

I was pleasantly surprised by the improved clarity of the paper. There are nevertheless a few minor improvements I can suggest.

1. The caption to Fig. 1 ends with a reference to "theories that allow a small amount of signaling." I don't know what the authors mean or whether it could even make sense. Certainly in quantum mechanics, any signaling arising from correlations would inevitably violate relativity. The authors should either explain what they have in mind or leave this reference out.

By a "small amount of signalling", we mean that the no-signalling condition, which is stated in equation (10) of our manuscript, only needs to be satisfied approximately. This can be expressed using two inequalities, and can therefore be seamlessly integrated into our linear programming approach. We agree that a small amount of signalling would violate relativity, which may be deemed unphysical. However, we merely see this as an example for what kind of constraints can be expressed in our framework, and as an illustration of the generality of our approach.

To clarify this point, we changed the corresponding sentence from

The latter assumption could be relaxed further to theories that allow a small amount of signalling.

to

The latter assumption could be relaxed further to theories that allow a small amount of signalling, in the sense that the no-signalling equation (10) is only satisfied approximately.

Next, he or she writes:

2. I don't know what the notation $E = E_1 E_2$ just above Eq. (2) means.

Indeed we forgot to mention what we mean by this equation. We use this notation to indicate that system E is subdivided into two subsystems E_1 and E_2 . We changed that part of the sentence from

where $A' = B$ and $E = E_1 E_2$

to

where $A' = B$ and where system E is subdivided into subsystems $E = E_1 E_2$

in order to make this clear.

The reviewer's next comment is as follows:

3. The caption to Fig. 2 mentions E_{in} which is nowhere defined (as far as I can see). Better use the notation of the text.

The distinction between E and E_{in} is indeed a subtle one. The distinction is necessary because the input system of the environment does not need to have the same dimension as the output system of the environment. However, we agree that it is not worth making this distinction in the main article. It is thoroughly explained in the Supplementary Information. We removed the subscript "in" in the main article, which leaves the points that we make untouched.

Moreover, he or she writes:

Also, the caption mentions "The channel's ability to preserve quantum information - that is, the amount of decoherence", which seems backwards. Doesn't the word "coherence" parallel "quantum information" better than "decoherence"?

We thank Reviewer #1 for pointing out this potential source of confusion. We changed the according sentence from

The channel's ability to preserve quantum information - that is, the amount of decoherence - can be characterized by how well it preserves entanglement between an outside system SA and SA' .

to

The channel's (in)ability to preserve quantum information - and therefore the amount of decoherence - can be characterized by how well it preserves entanglement between an outside system SA and SA' .

The last issue raised by Reviewer #1 is as follows:

4. Similarly, the caption to Fig. 4 states that "decoherence on a single quantum system can be understood fully by how well the decoherence process preserves entanglement", which is also backwards. Decoherence does not preserve entanglement, so I suggest "destroys" or a synonym instead of "preserves".

Again, we the reviewer for pointing this out. We changed that sentence from

... decoherence on a single quantum system can be understood fully by how well the decoherence process preserves entanglement ...

to

... decoherence on a single quantum system can be understood fully as the process' inability to preserve entanglement ...

Response to Reviewer #2

Reviewer #2 writes:

The authors made a number of important changes in presentation, both better explaining the motivation and carefully explaining (or, at least, defining) the relevant quantities and subsystems.

My concerns about validity of the proposed upper bounds are resolved, and I have no further objections about the technical details and the presentation.

Regarding the significance of this work, after reading the authors response and reading of the improved version of the manuscript, I am convinced that it deals with an important question (how to deal with potential consequences of quantum gravity even if we might to make changes to quantum mechanics). The authors provide a very useful and inspiring insight on this issue, and I recommend to publish the manuscript in its present form.

We thank Reviewer #2 for this positive assessment. As he or she recommends publication of the manuscript in its present form, we assume that all of the concerns have been addressed and that no further changes need to be made.

Response to Reviewer #3

Reviewer #3 writes:

I find that the refereeing process is improving the manuscript, also thanks to recommendations by other referees. This is a good interesting paper that will surely influence rather strongly the community interested in such studies, but I remain skeptical that it will have appeal extending beyond that community. This broader appeal would have been achieved for example if the authors had provided compelling arguments essentially explaining why for this next revolution it should be so important to have the most appropriate tools for interpretation of data. I do not find such a compelling case in the manuscript or in the reply, and I would still take as reference the fact that for all previous major "physics revolutions" we (the physicists of those times) did just fine employing rather inadequate tools of data analysis (like using classical-mechanics-optimized data analysis for many analyses that led to the discovery of quantum mechanics).

We thank the reviewer for calling our manuscript a good and interesting paper and for the assessment that it will strongly influence the

community. We are pleased to take notice that Reviewer #3 considers our approach to be an “appropriate tool” for the interpretation of data.

We are not sure what kind of arguments the reviewer is expecting as to why appropriate tools for the interpretation of data are important. In our opinion, this is what physics is all about: devising models for a consistent interpretation of the quantifiable observations (that is, data) that we make about nature. Our approach provides new tools for testing a very general class of such models. Respectfully, we would like to point out that we are highly sceptical about whether the alleged sufficiency of “inadequate” tools in the past should lead us to the conclusion that the development of new tools is unnecessary. We strongly believe that the development of new tools and approaches to long-standing problems is crucial to resolve them. What, if not new approaches for example for the interpretation of data, can resolve the problems that current tools are unable to solve?